# Optimization of an Ultrasound-Assisted Extraction Condition for Flavonoid Compounds from Cocoa Shells (*Theobroma cacao*) Using Response Surface Methodology

**DOI:** 10.3390/molecules24040711

**Published:** 2019-02-16

**Authors:** Arief Huzaimi Md Yusof, Siti Salwa Abd Gani, Uswatun Hasanah Zaidan, Mohd Izuan Effendi Halmi, Badrul Hisyam Zainudin

**Affiliations:** 1Malaysia Cocoa Board, Cocoa Innovative and Technology Centre, Lot 12621 Nilai Industrial Area, 71800 Nilai, Negeri Sembilan, Malaysia; ariefhuzaimi@koko.gov.my (A.H.M.Y.); badrul@koko.gov.my (B.H.Z.); 2Halal Products Research Institute, University Putra Malaysia, Putra Infoport, 43400 Serdang, Selangor, Malaysia; 3Department of Agriculture Technology, Faculty of Agriculture, University Putra Malaysia, 43400 Serdang, Selangor, Malaysia; 4Department of Biochemistry, Faculty of Biotechnology and Biomolecular Sciences, University Putra Malaysia, 43400 Serdang, Selangor, Malaysia; uswatun@upm.edu.my; 5Department of Land Management, Faculty of Agriculture, University Putra Malaysia, University Putra Malaysia, 43400 Serdang, Selangor, Malaysia; m_izuaneffendi@upm.edu.my

**Keywords:** cocoa shell, ultrasound-assisted extraction, response surface methodology, flavonoids

## Abstract

This study investigates the ultrasound-assisted extraction of flavonoids from Malaysian cocoa shell extracts, and optimization using response surface methodology. There are three variables involved in this study, namely: ethanol concentration (70–90 *v*/*v* %), temperature (45–65 °C), and ultrasound irradiation time (30–60 min). All of the data were collected and analyzed for variance (ANOVA). The coefficient of determination (R^2^) and the model was significant in interaction between all variables (98% and *p* < 0.0001, respectively). In addition, the lack of fit test for the model was not of significance, with *p* > 0.0684. The ethanol concentration, temperature, and ultrasound irradiation time that yielded the maximum value of the total flavonoid content (TFC; 7.47 mg RE/g dried weight (DW)) was 80%, 55 °C, and 45 min, respectively. The optimum value from the validation of the experimental TFC was 7.23 ± 0.15 mg of rutin, equivalent per gram of extract with ethanol concentration, temperature, and ultrasound irradiation time values of 74.20%, 49.99 °C, and 42.82 min, respectively. While the modelled equation fits the data, the T-test is not significant, suggesting that the experimental values agree with those predicted by the response surface methodology models.

## 1. Introduction

Waste from the cocoa industry, such as cocoa shells and cocoa pod husk, is usually discarded during the processing of cocoa (*Theobroma cacao* L.). Cocoa powder is the main ingredient of chocolate. Approximately 75% of the cocoa fruit is wasted during this process. To reduce this amount of waste and to increase the value of the waste products, several strategies have been investigated. Abundant of waste from this industry can be turned into a value-added product. Usually, this by-product of cocoa production is used as boiler fuel [1]. Other applications of the cocoa shell include particle board making [2], dietary fiber [3], fertilizer and animal feed [4], and activated carbon [5]. Additionally, there are many studies in vitro and in vivo on cocoa polyphenols and on its antioxidant capacity, including cocoa pulp [6], cocoa butter [7], cocoa pod husk [8], and so on [9,10,11]. The majority of those studies show that cocoa polyphenols provide notable health benefits because of their polyphenol and flavonoid monomer concentrations [12].

The cocoa shell composition is quite variable and is dependent on many factors, such as the origin, clone, processing, and pre-processing (along with the process line from seed to plant and from plant to product). Each of these stages yields a different chemical composition [13]. For a comparison of the fat content, the physical constants of the fat in the cocoa shell were similar to those in the cocoa butter from cocoa nibs [14]. However, within the same process, the chemical constants in the cocoa butter were lower than the fat content of the cocoa shell [14]. Additional studies show that the polyphenols level found in the cocoa shell are different [15,16]. However, in those studies, no specific composition of the type polyphenols was carried out.

Polyphenols are a product from a second metabolism plant, and are characterized as an aromatic compound, possessing an aromatic ring of one or more hydroxyl substituents. Flavonoids are one type of polyphenols; there are many types of flavonoids, including flavones, isoflavonoids, flavonol, and flavanones [17]. The major phenolic compounds found in cocoa are flavanols [18]. A previous study stated that the total flavonoid content (TFC) in the cocoa shell and pod husk for different kinds of clones have a different flavonoid yield [19]. All TFCs that are obtained are different because of the previously mentioned different conditions.

In order to extract polyphenols and flavonoids, a proper technique should be chosen, as different techniques yield different amounts of the desired biochemical [20]. A traditional technique usually yields a low amount of extracted flavonoids, as compared to modern technologies, such as microwave-assisted extraction (MAE) [21], ultrasound-assisted extraction (UAE) [22], and supercritical fluid extraction (SFE) [23]. Apart from the type of process used to extract the polyphenols, other factors, such as the type of solvent, duration of the extraction, and temperature of the extraction play an important role in achieving the optimum extraction of polyphenols from the plant [24]. 

The principle of UAE is based on the fundamentals of wave frequency, different than that of the fundamentals of sound. Generally, sound waves are classified into three categories, which are audible waves (10 Hz–20 kHz), infrasonic waves (<16 Hz), and ultrasonic waves (between 20 kHz and 10 MHz) [25]. The catories above 10 MHz are known as microwaves. Usually, the power use in UAE is in between 50–400 W. During the UAE process, the microbubbles were generated because of the different amplitude of ultrasonic waves [26]. As a result of the continuous changes of amplitude, the cavitation bubbles oscillate and collapse, creating several physical effects. Some physical effects include shock waves, microjets, turbulence, and shear force, as a result of improving the mass and heat transfer through the disruption of the cell walls, and increasing the pores on the surfaces of the cells, leading to the improved release of the target bioactive compounds from the natural sources [27]. 

Therefore, in order to obtain the optimum process needed to extract the flavonoid, the optimization technique using response surface methodology (RSM) was chosen for this study. The RSM technique involves planning and performing a set of experiments based on the number of input variables [28]. The complexity of the interactions between the variables will be analyzed until a valid model is obtained [28]. The resultant model contains all of the information on the effect of the experimental conditions, and limits the experimental run [29]. In addition, the model can be used to predict future observations within the model ranges. 

The main objective of this study was to obtain the highest flavonoid level by optimizing the extraction condition for Malaysian cocoa shell extracts (MCSE). The three variables were the ethanol concentration (70–90% *v*/*v*), temperature (45–65 °C), and ultrasound irradiation time (30–60 min).

## 2. Material and Methods

### 2.1. Materials and Chemicals

The cocoa fruit was purchased from Pusat Penyelidikan dan Pembangunan Koko, Jengka, Pahang, Malaysia. The fruit was cut in order to take the fresh bean out. All of the fresh beans were freeze-dried to remove all of the enzymes and pulp juice. Next, the Malaysian cocoa shell (MCS) was manually removed from the bean and pulp. The other chemicals used in this study are rutin, aluminum chloride, 2,2-diphenyl-2-picrylhydrazyl (DPPH), ascorbic acid, Tris–HCl buffer, 2,4,6-Tri(2-pyridyl)-s-triazine (TPTZ), Ferric chloride hexahydrate, acetate buffer, Ferrous sulfate heptahydrate, β-Carotene, chlorofom, linoleic acid, Tween 20, procyanidin standard, epicatechin standard, and ethanol. All of the chemicals were of analytical grade. 

### 2.2. Ultrasound-Assisted Extraction

First, MCS was crushed in a mechanical blender (IKA, Staufen, Germany). Next, 1 g of MCS was mixed with a 50 mL solvent, and was extracted using a sonication bath machine (Wiseclean 40 kHz, 296 W, Wonju-si, Korea). All of the extraction conditions are described in Table 1. Next, the aqueous extract was filtered using Watman filter paper number 4, and the solvent was removed using a rotary evaporator (IKA, Staufen, Germany) and was freeze-dried (Labconco, Kansas City, MO, USA) to get the crude extract, and was labelled as Malaysian cocoa shell extract (MSCE). The crude extract was stored at −40 °C in storage vials until further analysis.

### 2.3. RSM Design and Statistical Analysis

A central composite design was used to determine the optimal condition for ultrasound-assisted extraction. The variables were set as X_1,_ X_2_, and X_3_ for the ethanol concentration (%), temperature (°C), and ultrasound irradiation time (minutes), respectively. The five levels (−α, −1, 0, 1, α), including alpha (α), were coded (Table 1), and 20 experimental runs were created as a base on the central composite design (CCD) (Table 2). The total flavonoid content was considered as the independent variable (Y), and the data were analyzed using the second polynomial equation (Equation (1)). The interaction variables, both linear and quadratic, were analyzed using an analysis of variance (ANOVA) to determine the coefficient of the second polynomial equation, lack of fit, and the coefficient of determination (R^2^) on the total flavonoid content. All of the experimental runs were run in triplicate and the data were analyzed using Expert Design Software (version 10, Stat-Ease Inc, Minneapolis, MN, USA).
(1)$$Y=\beta_{0}+\overset{3}{\underset{i=0}{\sum }}\beta_{i}X_{i}+\overset{3}{\underset{i=0}{\sum }}\beta_{ii}X_{ii}^{2}+\overset{3}{\underset{i\neq j=0}{\sum }}\beta_{ij}X_{i}X_{j}$$
where *Y* is the response variable; $\beta_{0}$ is a constant; and $\beta_{i}$, $\beta_{ii}$, and $\beta_{ij}$ are the linear, quadratic, and interactive coefficients, respectively. $X_{i}$ and $X_{j}$ are the levels of the independent variables.

### 2.4. Determination of TFC

The TFC in MCS was determined using a procedure by Chang C.C. [30], with several modifications. For each sample, 2% aluminum chloride (1 mL) was mixed with 1 mL of 500 µg/mL of the sample. Then, the mixture was vortexed for 10 s and incubated for 20 min. Next, the absorbance was measured at 507 nm using a UV-Visible Spectrophotometer (Cary 60, Agilent Technologies, Santa Clara, CA, USA). All of the samples were analyzed in triplicate, and the results were recorded (Table 2). The TFC was expressed as mg rutin equivalents per gram of dried weight (mg RE/g DW), using a calibration curve constructed with rutin (0–1000 µg/mL) with R^2^ = 0.9989.

### 2.5. Ramp Function and T-Test

The polynomial formula was validated using a ramp function graph experiment, and T-tests were performed for the data collected from this experiment. Three random solutions were created from the polynomial design model, with different conditions from the design RSM, which were tested for their predictions and experimental values (Table 3). The desirability was fit to a value of 1.00. All of the experimental simulations were run in triplicate, and the data collected from the experiment ramp function were tested for statistical data using the T-test from the Minitab Software (version 14, Minitab, Pennsylvania, PA, USA).

### 2.6. Identification of Flavonoids Compounds

Ultra-high-performance liquid chromatography (UHPLC) was carried out on the ACQUITY UPLC I-Class system (Waters, Manchester, UK), consisting of a binary pump, a vacuum degasser, an autosampler, and a column oven. The flavonoid compounds were chromatographically separated using a column (ACQUITY UPLC HSS T3, 100 mm × 2.1 mm × 1.8 μm, Waters, Manchester, UK), maintained at 40 °C. A linear binary gradient of first water (0.1% formic acid), and then acetonitrile, was used for mobile phases A and B, respectively. The mobile phase composition was altered during the run as follows: 0 min, 1% B and 99% A; 0.5 min, 1% B and 99% A; 16.00 min, 35% B and 65% A; 18.00 min, 100% B and 0% A; and 20.00 min, 1% B and 99% A. The flow rate was set to 0.6 mL/min with a 1 µL injection volume. The data were acquired in the high-definition mass spectrometry elevated energy (HDMS^E^) mode in the range *m*/*z* 50–1500 at 0.1 s/scan. Thus, two independent scans with different collision energies (CE) were alternatively acquired during the run of a low-energy (LE) scan at a fixed CE of 4 eV, and a high-energy (HE) scan, where the CE was ramped from 10 to 40 eV. Ultra-pure argon (99.999%) was used as the collision-induced-dissociation (CID) gas.

### 2.7. Determination of Antioxidant Activities

The antioxidant activities were evaluated using a 2,2-diphenyl-2-picrylhydrazyl (DPPH) radical scavenging assay, Ferric reducing/antioxidant power (FRAP) assay, and β-Carotene–linoleate bleaching assay (BCB). All of the method preparations and calculations of DPPH, FRAP, and BCB were adopted with some modifications, and their absorbance was read using UV-Visible Spectrophotometer (Cary 60, Agilent Technologies, Santa Clara, CA, USA). Briefly, the methods from Azizah Othman et al. (2007) [31] were used to determine the scavenging activity from MSCE. Then, 200 µL of cocoa extract (5000 µg/mL) or ascorbic acid was mixed with 800 µL Tris-HCl buffer (100 mM, pH 7.4) and 1 mL of 500 µM DPPH. The mixture was vortexed for 2 s and left to stand for 20 min at room temperature in a dark room. The absorbance was read at 517 nm. The scavenging effect on of the DPPH radical was calculated using Equation (2), as follows:(2)$$Scavenging\;effect=\left(1-\frac{absorbance\;sample\;at\;517\;nm}{absorbance\;control\;at\;517\;nm}\right)\times 100$$

For the FRAP assay, the antioxidant power was evaluated using the method described by José R. et al. (2012) [32]. The FRAP reagent was prepared by mixing 300 mM of acetate buffer (pH 3.6), 10 mM TPTZ, and 20 mM FeCl_3_. 6H_2_O in a ratio of 10:1:1, at to 37 °C. Then, 3 mL of the FRAP reagent was mixed with 100 µL of the sample or calibration standard. The mixtures were then incubated at 37 °C for 10 min. Each sample was run in triplicate. The absorbance was read at 593 nm. The FRAP value was calculated using a calibration curve constructed from FeSO_4_·7H_2_O (7.8–1000 µg/mL), with R^2^ = 0.9928. For the β-carotene–linoleate bleaching assay, the antioxidant activity of MSCE was measured based on the b-carotene bleaching method described by Azila Karim et al. (2014) [33]. Then, 2 mg of β-carotene (dissolve in 0.2 mL chlorofom) was mixed with 0.2 mL of linoleic acid, 2 mL Tween 20, and 100 mL of distilled water, to make a β-carotene solution. Then, 2 mL of the test solution was pipetted into a vial and immediately mixed with 200 µL of MSCE, and incubated for 2 h at 50 °C. The absorbance was read at 470 nm. The degradation rate (DR) was calculated according to the first order kinetics, using Equation (3), as follows:(3)$$\ln\frac{a}{b}\times \frac{1}{t}=Degration\;rate_{sample\;or\;standard}$$
where ln is a natural log, *a* and *t* are the initial absorbance (470 nm) at time 0, and *b* is the absorbance (470 nm) at 120 min. Antioxidant activity was expressed as percent of inhibition relative to the control, using the following Equation (4):(4)$$Antioxidant\;activity\;\left(\%\right)=\left(\frac{Degration\;rate_{control}-Degration\;rate_{sample\;or\;standard}}{Degration\;rate_{control}}\right)\times 100$$

### 2.8. Statistical Analysis

The data are expressed as the mean ± standard error. The significance was evaluated using analysis of variance testing (ANOVA) on Expert Design Software, and the T-test for validation was performed on Minitab Software. 

## 3. Results and Discussion

### 3.1. Fitting the Model

All of the data recorded from the experiments were analyzed using analysis of variance (ANOVA) tests to evaluate the significance and fit of the model, through determining the F-value and *p*-value (Table 4 and Table 5). The model was significant (*p* < 0.0001; F-value = 56.54). On the other hand, the coefficient of determination (R^2^) and the lack-of-fit test of the model were 0.9759 and 0.0684, respectively. This shows that the second-order model equation generated by the software (Equation (5)) can be used to predict future observations within the design range.
(5)$$Y_{TFC}=-29.87328+0.61629X_{1}+0.37769X_{2}+0.13328X_{3}-0.000447797X_{1}X_{2}-0.000148467X_{1}X_{3}-0.00103289X_{2}X_{3}-0.00375889X_{1}^{2}-0.00271499X_{2}^{2}-0.00072126X_{3}^{2}$$

### 3.2. Respond Surface Analysis of TFC

All of the data are presented in Figure 1, Figure 2 and Figure 3. Figure 1 shows the response surface plot and contour plot of the ethanol concentration (X_1_) and temperature (X_2_). The ethanol concentration is the most important parameter affecting the TFC value. From the experiment, the TFC value increased along with the ethanol concentration, and started to decrease after a concentration of 80%. Using a solvent mixture could increase the amount of flavonoid or the other compound composition extract from both ends of the polarity (highest and low polarity compound) [34]. By applying a high temperature during the extraction process, the polarity of the solvent can be decreased, and can be suitable for the targeted compound, as the solvent mixture (water-solvent) could improve the effectiveness of the extraction and increase surface area for the solvent solid contact [35]. The same effect was observed when the temperature was increased and started to decrease after 55 °C. The higher temperatures decrease the flavonoid content as flavonoids are sensitive to high temperatures. High temperatures and longer exposure times can reduce the diversity of the extracted polyphenols; the only phenolic compound whose yield increased with temperature was likely a product of the thermal degradation of the polyphenols [32]. A similar result was obtained by Azahar et al. (2017) [36], where the total flavonoid of *Curcuma zedoaria* leaves was optimized at 60 °C; the TFC decreased as the temperature continued to increase.

The increased length of the cocoa shell treatment decreased the TFC value (Figure 2 and Figure 3). The optimum time to extract the TFC from the cocoa shell is 45 min. The degradation of TFC is as a result of the extended treatment on the cocoa shell. In agreement, previous studies suggest that a more prolonged treatment lowers the concentration of polyphenols, because of their degradation [32]. Moreover, the result is similar to that in Liu et al. (2013) [37], where the yield of a phenolic compound decreased upon a longer treatment. The decreased TFC value by a longer treatment was probably due to the mechanism of ultrasound. A previous study was done by Suslick (1990) [38] on sonochemistry; there were four variations of cavitation (acoustic, hydrodynamic, optic, and particle) involved during the ultrasound irradiation. During the process, only the acoustic and hydrodynamic cavitation could produce the intensities needed to induce the physical or chemical changes [39]. Even though the chemical influences of the ultrasound may not happen through a direct interaction with the molecular species the cavitation occurrence (formation, growth, and implosive microbubble or collapse cavities) may release a large amount of highly localized energy that can change the chemical in the system [40]. 

Furthermore, free radicals are formed during the acoustic cavitation, and at an ambient temperature, these radicals could be utilized for the acceleration of chemical reactions. Another factor, such as local turbulence and acoustic streaming (liquid micro-circulation), might contribute to decreasing the mass transfer resistance in the system [40]. Therefore, ultrasound irradiation can be considered as a combination of a chemical reaction, by utilizing the factor mentioned that is associated with an increase in the temperature [41]. All of the factors mentioned before are probable reasons for why the TFC value decreases when it reaches an optimal value.

UAE offers several advantages compared to conventional extraction, for example, simplified manipulation, high reproducibility in shorter times, reduced solvent consumption, and lower energy consumption. For comparison, a study was done by Lian He et al. (2018) [42] on the extraction of polysaccharides from the *Dendrobium officinale* stem, which shows that UAE (20 kHz, at 400 W) gives the highest yield of 20.55% polysaccharide, compared to the MAE (600 W) 17.74% and hot water extraction 14.77% polysaccharide. The other study was done by Carla and Andrea (2018) [43] on the extraction of polyphenols from saffron floral using MAE (800 W) and UAE (26 kHz, 200 W), which obtained a similar result. By using UAE, the polyphenols recovered from saffron the floral were 4016 mg GAE 100 g DM^−1^ compared to the MAE 3108 mg GAE 100 h DM^−1^ [43]. A massive difference between the method of extraction showed that another method should increase the duration of the treatment (hence increase energy consumption) or increase the solvent consumption, in order to obtain a similar result as extraction using UAE. Thus, the UAE can be optimized by using less energy consumption, shortening the duration of the extraction process, and decreasing the solvent consumption. 

### 3.3. Verification of the Model

This study aimed to verify the model that can be used with any set of conditions within the design range. Using the desirability function approach for cocoa shells by choosing lower temperatures, concentrations, and ultrasound irradiation times with the desirability value yields a final result of 1.00 for the simultaneous optimization. The T-test shows that the data is not significant (*p* > 0.05) between the predicted and experimental values (Table 6). Based on the results, the predicted and experimental values were comparable. Therefore, the extraction conditions obtained by the RSM model are satisfactory.

### 3.4. Identification of Flavonoid Compound

The base peak chromatogram of the flavonoid cocoa shell extract obtained by ultra-high-performance liquid chromatography-quadrupole time-of-flight mass spectrometry (UPHLC-QTOF-MS) and the characterized compounds are presented in Table 7, and are numbered according to their retention times. All of the compounds were characterized through the interpretation of their mass spectra (QTOF mass analyzer in negative mode), while considering the information provided by the literature and databases. All of the flavonoid compounds that were found were then labelled as tentative or confirmed. The tentative label refers to a compound only identified from the Waters library with the accepting an error below 5 ppm and a theoretical fragment more than 1 [44], while the confirmed label refers to the compound identified from the Waters Library compared with the standard of the same compound. Because of the lack of standards, only epicatechin and procyanidin B2 were validated using standards. The MSCE negative ion mode product of both of the compounds (chromatogram mass spectrum, low energy of mass spectrum, and high energy of mass spectrum) are shown in Figure 4 and Figure 5. Previous studies report that epicatechin is found in fermented cocoa bean [44], cocoa pod husk [33], cocoa powder [45], and cocoa nibs [46]. Procyanidin trimer, procyanidin tetramer, procyanidin pentamer procyanidin B2, procyanidin C1, procyanidin B4, and procyanidin A2 are types of oligomers of flavonoids, and are found in the fermented and unfermented cocoa beans [47]. On the other hand, the process of extraction and the type of cocoa bean will give a different type of flavonoid and flavonoid yield. For comparison, procyanidin B and B2 were found in the different types of cocoa powder, nibs, and nibs powder, while procyaidin C is only found in the cocoa extract using sonication, but is not found in the pressurizes hot water extraction [46].

### 3.5. Antioxidant Activities of MSCE

Three antioxidant assays were used to evaluate the antioxidant activities on MSCE, with a higher TFC value S7 and R1 selected from the RSM model and ramp function as in Table 8. Both methods (DPPH and FRAP) were compared with the acid ascorbic, (AA) and the β-carotene bleaching assay was compared with the BHT. In the scavenging activity assay, the proton radical scavenging activity is known to be one of several mechanisms for measuring antioxidant activity. DPPH contains proton free radicals, and its absorbance can be read at 517 nm. Generally, when DPPH encounters radical proton scavengers, its purple color fades, and change to a yellowish color to show the presence of an antioxidant in the sample [48]. 

The FRAP assay measures the reducing potential of an antioxidant reacting with a ferric tripyridyltriazine complex, and produces colored ferrous tripyridyltriazine (blue color) at a low pH [49,50]. The presence of the compounds can be tracked by measuring the increase in the absorbance value as the compound starts breaking the free radical by donating a hydrogen atom. The previous study shows that the FRAP assay can be used to measure the mechanism of the Fenton reaction by chelating a metal ion such, as Fe^2+^, which converts hydrogen peroxide to the hydroxyl radical [51]. 

In the β-carotene bleaching assay, linoleic acid was used to create the hydroperoxides compound as free radicals during the incubation. This free radical was used to evaluate the antioxidant activity of MSCE, to maintain the yellow color of the test solution after 120 min [52]. The antioxidant activity from S7 is lower than acid ascorbic and BHT, but similar to the R1. The similarity is probably because the total flavonoid contents are similar, even though the parameter of the extraction condition is different. The present of oligomer procyanidin, as in Table 7, should yield a high scavenging activity compared to the monomer, such as epicatechin. A previous study was done by Vennat et al. (1994) [53], which showed that monomer procyanidin is less effective against superoxide anions, compared to the dimers and trimes procyanidin, while the heptramer and hexamers demonstrate a greater superoxide scavenging activity than tetramer [53]. 

Even though there is evidence of the effectiveness of oligomer procyanidin, the antioxidant activity was influenced by other factors, such antioxidant concentration, method of extraction, extraction medium, particle size of natural sources, temperature, Ph of medium [54], chemical structures, and position in the molecule [55]. The oligomer procyanidin is also present in the cocoa bean, cocoa powder, and cocoa pod husk, as mentioned before. The percentage of oligomer procyanidin is probably higher because of the extraction method involved in the ultrasound. The previous study shows that an ultrasound can increase the procyanidin content in the extract compared with another method. The method extraction on *Larix gmelinii* was compared between an 80% solvent (ethanol) and ionic liquid, such as Bmim-Br (1-butyl-3-methylimidazolium bromide) [56]. It was found that more 98% of the procyanidins content was extracted during the first 30 min of UAE, and the result did not increase any further after 30 min as the application of the ultrasound continued. 

We conducted a kinetic study (data not published) for S7 (concentration between 1000–8000 µg/mL) at the same wavelength used on antioxidant method. The S7 antioxidants (DPPH and FRAP) are increased as the concentration increases with a more than 50% scavenging effect and one µMoles/L, respectively. For the BCB assay, the MSCE S7 and R1 had a similar antioxidant activity to BHT, as the concentration increased.

## 4. Conclusions

UAE was used in the extraction of the flavonoid compounds from MCSE, and was optimized by using the center composite design for the response surface methodology. The CCD was used to evaluate the complexity of the variables (ethanol concentration, temperature of extraction, and ultrasound irradiation time) to extract the flavonoids from the MCS. From the experiment, the results determined that the variables are significant only with respect to the ethanol concentration. However, the interaction between the variables in the model are significant (*p* < 0.001), allowing for the incorporation of a quadratic model. From the T-test of the ramp function, the generated polynomial model can be repeated at any point within the design range, and the results from the UHPLC-QTOF-MS show that there are nine compounds of flavonoids found in the MCSE, and two of them were validated with the standard. The optimization conditions are as follows: ethanol concentration of 80%, temperature of 55 °C, and an ultrasound irradiation time 45 min. Under these conditions, the experimental and model-predicted TFC values are virtually the same (7.47 ± 0.11 mgRE/g DW and 7.41 mgRE/g DW, respectively). Therefore, in this study, MCSE was successfully optimized.

## Figures and Tables

**Figure 1 molecules-24-00711-f001:**
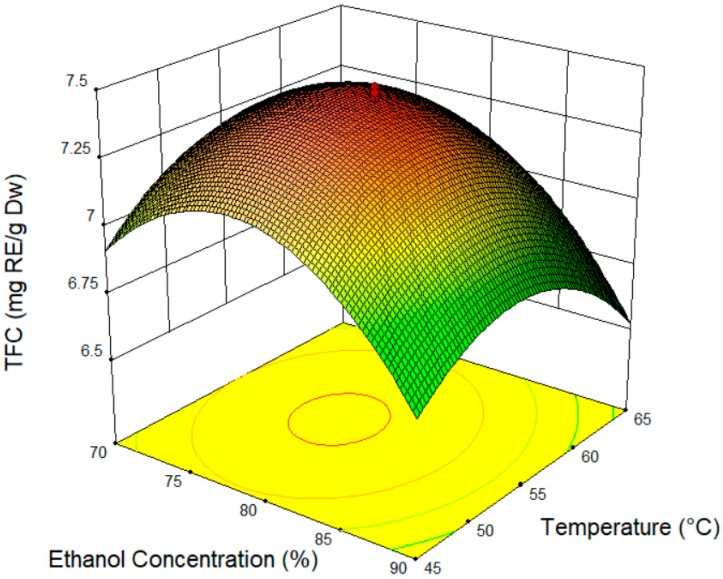
Response surface plot and of ethanol concentration (X_1_) and temperature (X_2_).

**Figure 2 molecules-24-00711-f002:**
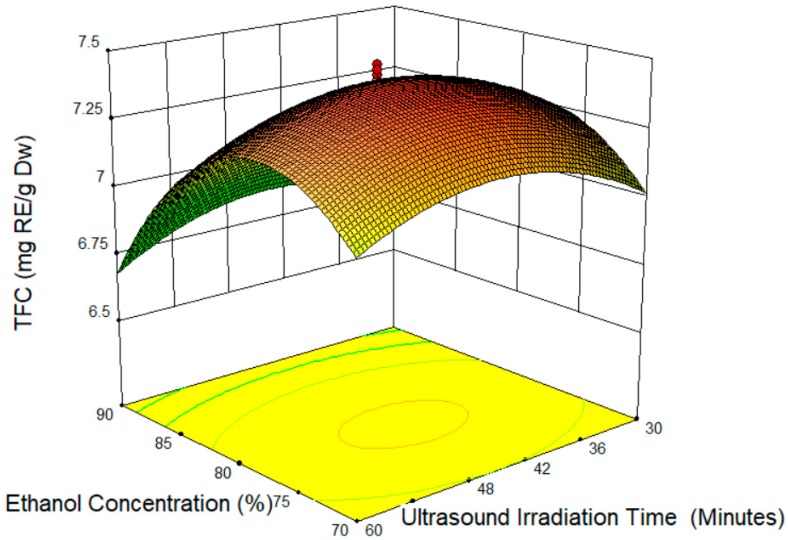
Response surface plot of ethanol concentration (X_1_) and ultrasound irradiation time (X_3_).

**Figure 3 molecules-24-00711-f003:**
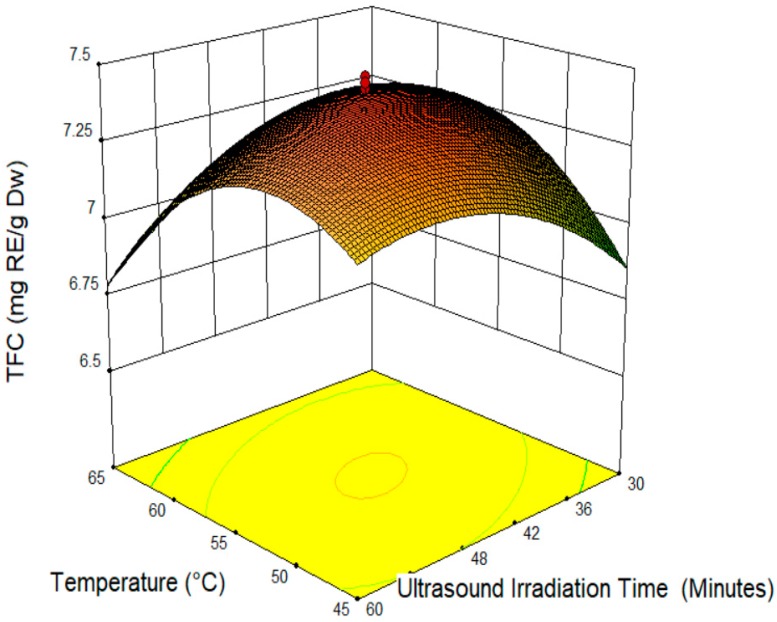
Response surface plot of temperature (X_2_) and ultrasound irradiation time (X_3_).

**Figure 4 molecules-24-00711-f004:**
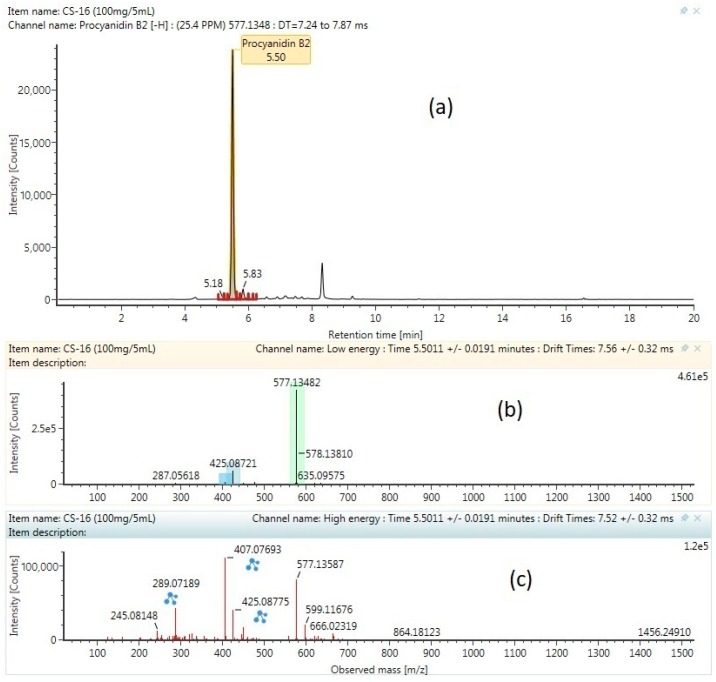
MSCE negative ion mode product of procyanidin B2 compound: (**a**) chromatogram mass spectrum; (**b**) Low energy of mass spectrum; (**c**) high energy of mass spectrum.

**Figure 5 molecules-24-00711-f005:**
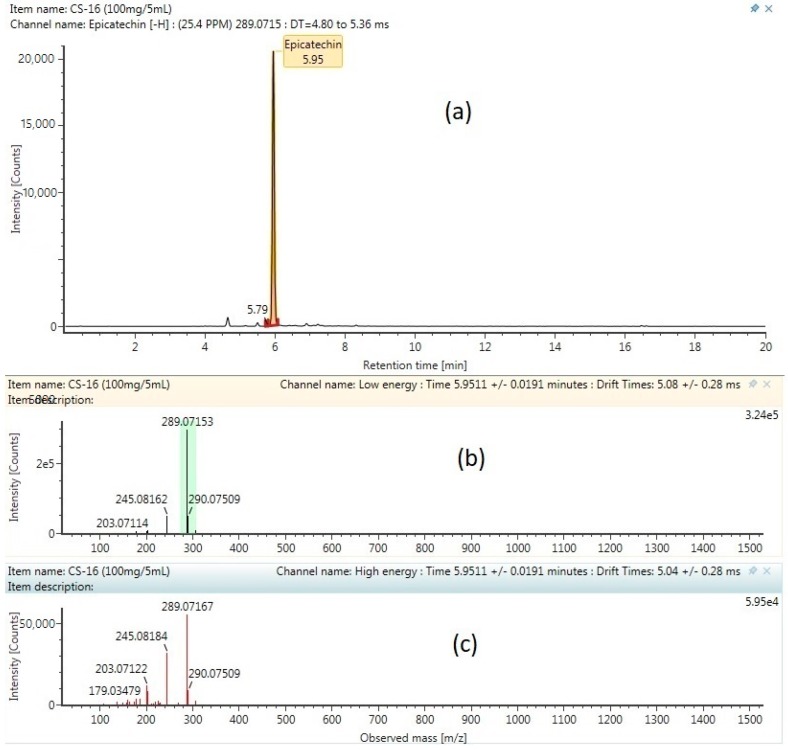
Malaysian cocoa shell extract (MSCE) negative ion mode of epicatechin compound: (**a**) chromatogram mass spectrum; (**b**) low energy of mass spectrum; (**c**) high energy of mass spectrum.

**Table 1 molecules-24-00711-t001:** Independent variables and their levels for central composite design.

Independent Variables	Levels				
	−α	−1	0	1	α
Ethanol concentration (X_1_) (%)	63.18	70	80	90	96.82
Temperature (X_2_) (°C)	38.18	45	55	65	71.82
Ultrasound irradiation time (X_3_) (minutes)	19.77	30	45	60	70.23

**Table 2 molecules-24-00711-t002:** Central composite design of ethanol concentration, temperature, ultrasound irradiation time, and their observed responses using UAE. DW—dried weight; TFC—total flavonoid content.

Run	Actual Level			Coded Level			TFC Experiment (mg RE/g DW)	TFC Prediction (mg RE/g DW)
	Ethanol Concentration, X_1_ (%)	Temperature, X_2_ (°C)	Ultrasound Irradiation Time, X_3_ (min)	Ethanol Concentration, X_1_ (%)	Temperature, X_2_ (°C)	Ultrasound Irradiation Time, X_3_ (min)		
1	70.00	45.00	60.00	−1.00	−1.00	1.00	6.85 ± 0.05	6.92
2	70.00	65.00	60.00	−1.00	1.00	1.00	6.56 ± 0.30	6.64
3	80.00	38.18	45.00	0.00	−1.62	0.00	6.66 ± 0.31	6.69
4	96.82	55.00	45.00	−1.62	0.00	0.00	6.02 ± 0.32	6.06
5	70.00	65.00	30.00	−1.00	1.00	−1.00	6.84 ± 0.07	6.91
6	90.00	65.00	30.00	1.00	1.00	−1.00	6.57 ± 0.31	6.54
7	80.00	55.00	45.00	0.00	0.00	0.00	7.47 ± 0.11	7.41
8	80.00	55.00	45.00	0.00	0.00	0.00	7.43 ± 0.26	7.41
9	80.00	55.00	70.23	0.00	0.00	1.62	7.03 ± 0.02	6.94
10	80.00	71.82	45.00	0.00	1.62	0.00	6.68 ± 0.17	6.59
11	90.00	45.00	60.00	1.00	−1.00	1.00	6.67 ± 0.26	6.64
12	90.00	45.00	30.00	1.00	−1.00	1.00	6.43 ± 0.20	6.38
13	80.00	55.00	19.77	0.00	0.00	−1.62	6.92 ± 0.30	6.96
14	80.00	55.00	45.00	0.00	0.00	0.00	7.38 ± 0.22	7.41
15	90.00	65.00	60.00	1.00	1.00	1.00	6.12 ± 0.75	6.18
16	80.00	55.00	45.00	0.00	0.00	0.00	7.45 ± 0.36	7.41
17	70.00	45.00	30.00	−1.00	−1.00	−1.00	6.60 ± 0.10	6.58
18	80.00	55.00	45.00	0.00	0.00	0.00	7.32 ± 0.18	7.41
19	80.00	55.00	45.00	0.00	0.00	0.00	7.41 ± 0.33	7.41
20	63.18	55.00	45.00	−1.62	0.00	0.00	6.72 ± 0.25	6.62

**Table 3 molecules-24-00711-t003:** Condition for the ramp function graph.

Run	Ethanol Concentration, X_1_ (%)	Temperature, X_2_ (°C)	Ultrasound Irradiation Time, X_3_ (min)	TFC Experiment (mg RE/g DW)	TFC Prediction (mg RE/g DW)
1	74.20	49.99	42.82	7.23	7.30
2	80.60	43.34	41.17	6.92	7.01
3	77.53	58.79	19.77	7.08	7.02

**Table 4 molecules-24-00711-t004:** Analysis of variance for a fitted quadratic model of extraction of the flavonoid compound.

Source	Sum of Square	Degree of Freedom	Mean Square	F-Value	*p*-Value
Model	3.59	9	0.40	56.45	<0.0001
Residual	0.071	10	0.0071		
Lack-of-fit test	0.057	5	0.011	4.27	0.0684
Pure error	0.013	5	0.0027		
Cor. total	3.66	19			

R^2^ = 0.9807; R^2^_pred_ = 0.9633; Coefficient of variance (%) = 1.23; adequate precision = 22.542.

**Table 5 molecules-24-00711-t005:** Regression coefficient estimates and their significant tests for the quadratic model.

Source	Sum of Square	Degree of Freedom	Mean Square	F-Value	*p*-Value
X_1_	0.37	1	0.37	52.24	<0.0001
X_2_	0.015	1	0.015	2.05	0.1823
X_3_	0.00031	1	0.000312	0.044	0.8377
X_1_^2^	2.04	1	2.04	288.27	<0.0001
X_2_^2^	1.06	1	1.06	150.39	<0.0001
X_3_^2^	0.38	1	0.38	53.73	<0.0001
X_1_X_2_	0.016	1	0.016	2.27	0.1627
X_1_X_3_	0.00397	1	0.003968	0.56	0.4708
X_2_X_3_	0.19	1	0.19	27.19	0.0004

**Table 6 molecules-24-00711-t006:** T-test table of verification of the model.

Solution	Number of Samples	Mean	Standard Deviation	Standard Error Mean	95% Coefficient of the Interval	T-Value	*p*-Value
1	3	7.23	0.15	0.087	6.85, 7.61	0.00	1.000
2	3	6.81	0.35	0.204	6.04, 7.80	0.02	0.988
3	3	7.06	0.08	0.049	6.86, 7.29	−0.07	0.953

**Table 7 molecules-24-00711-t007:** Retention time, natural mass, observed mass-to-charge ratio (*m*/*z*), and identification status of the product ion of the flavonoid cocoa shell extract in negative ion mode.

Number	Component Name	Retention Time (min)	Natural Mass (Da)	Observed *m*/*z*	Identification Status and Label
1	Procyanidin B2	5.56	578.14243	577.1356	Identified, confirmed
2	Procyanidin trimer	5.57	866.20581	865.1991	Identified, tentative
3	Epicatechin	6.02	290.07904	289.0718	Identified, confirmed
4	Procyanidin C1	6.63	866.20581	865.1986	Identified, tentative
5	Procyanidin tetramer	6.95	1154.26920	1153.2606	Identified, tentative
6	Procyanidin trimer	7.15	866.20581	865.1981	Identified, tentative
7	Procyanidin pentamer	7.23	1442.33259	1441.3247	Identified, tentative
8	Procyanidin B4	8.36	578.14243	577.1346	Identified, tentative
9	Procyanidin A2	11.4	576.12678	575.1191	Identified, tentative

**Table 8 molecules-24-00711-t008:** Antioxidant activities of Malaysian cocoa shell extract (MSCE).

Antioxidant assay	S7	R1	Acid Ascorbic	BHT
a) Scavenging activity assay (%)	30.83 ± 1.55	29.56 ± 1.13	73.86 ± 2.26	Not detected
b) Ferric reducing/antioxidant power (µM Fe II g^−1^ extract)	319.20 ± 0.18	356.30 ± 0.24	816.05 ± 0.12	Not detected
c) β-Carotene–linoleate bleaching assay (%)	94.85 ± 0.39	93.79 ± 1.79	Not detected	99.87 ± 1.89

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
