# Peer review of "Optimization of an Ultrasound-Assisted Extraction Condition for Flavonoid Compounds from Cocoa Shells (Theobroma cacao) Using Response Surface Methodology"

_molecules, 2019, doi:10.3390/molecules24040711_

Round 1
Reviewer 1 Report
This article contribute to optimize conditions for flavonoid extraction from cocoa shell. Although the techniques are not new, their applications in re-using the waste cocoa shell can potentially benefit the cocoa industry.
The major issue of this article is that the authors made their major efforts on response surface methodology, and there are too much details in Table 1-5. The remaining part is skinny. More experiments and results are required to enrich the main body.
Much efforts required on academic writing, e.g.
l Use either US or UK English
l Line 51: levels?
l Line 56: states? Kinds?
l Line 58: have different flavonoid yield?
l Line 84: in analytical grade
l Line 121: what do you mean: “were tested for statistical data”
l Line 52: what do you mean: “the type polyphenols”
l Line 22: what do you: “significant extracction”;extraction level?
l Line 96: what do you: “were created a base on”
l Line 160: Figure 1 to 3
l Line 168: a similar result was obtained by
l Line 183: do you mean: is not sinificantly different between … ?
l Line 80: check the bracket
l The period is not necessary in subtitles
Author Response
Response to Reviewer 1 Comments
Point 1: The major issue of this article is that the authors made their major efforts on response surface methodology, and there are too much details in Table 1-5. The remaining part is skinny. More experiments and results are required to enrich the main body.
Response 1: The manuscript now includes the determination of antioxidant properties (2, 2-diphenyl-2-picrylhydrazyl radical scavenging assay, Ferric reducing/antioxidant power assay, and β-Carotene–linoleate bleaching assay) on MSCE with the sample higher TFC value (S7) selected from the RSM model. The data obtained were compared with the simulator run sample (R1), acid ascorbic and BHT. The method and discussion as below;
2.7 Determination of antioxidant activities.
Antioxidant activities were evaluated by using 2,2-diphenyl-2-picrylhydrazyl (DPPH) radical scavenging assay, Ferric reducing/antioxidant power (FRAP) assay, and β-Carotene–linoleate bleaching assay (BCB). All method preparation and calculation of DPPH, FRAP, and BCB were adopted with some modification, and their absorbance was read by using UV visible spectrophotometer (Cary 60 Agilent, USA). Briefly, the methods from Azizah Othman et al. (2007) [31] were used to determined scavenging activity from MSCE. 200µL of cocoa extract (5000µg/ml) or ascorbic acid was mixed with 800 µL Tris–HCl buffer (100 mM, pH 7.4) and 1 mL of 500 µM DPPH. The mixture was vortex for 2 seconds and left to stand for 20 minutes at room temperature in a dark room. Absorbance was read at 517 nm. The scavenging effect on of the DPPH radical was calculated using the following equation:
For FRAP assay, the antioxidant power was evaluated by using the method described by José R et al (2012) [32]. The FRAP reagent was prepared by mixing 300 mM acetate buffer (pH 3.6), 10 mM TPTZ and 20 mM FeCl3. 6H2O in a ratio of 10:1:1, at to 37 °C. Then 3 ml FRAP reagent was mixed with 100 µL of sample or calibration standard. The mixtured were then incubated at 37 °C for 10 minutes. Each sample was run in triplicate. The absorbance was read at 593 nm. The FRAP value was calculated using ca alibration curve constructed from FeSO4.7H2O (7.8 – 1000 µg/ml) with R2 = 0.9928. For the β-Carotene–linoleate bleaching assay, the antioxidant activity of MSCE was measured based on the b-carotene bleaching method described by Azila Karim et al. (2014) [33]. 2mg of β-Carotene ( dissolve in 0.2 ml chlorofom) were mixed with 0.2 ml of linoleic acid, 2 ml Tween 20 and 100 mL distilled water to make a β-Carotene solution. 2 mL of test solution was pipette into a vial and immediatelly mixed with 200 µl of MSCE and incubate for 2 hours at 50 ºC. The Absorbance were read at 470 nm. Degradation rate (DR) was calculated according to first order kinetics, using the following equation:
Where ln is a natural log, a and t is the initial absorbance (470 nm) at time 0, b is the absorbance (470 nm) at 120 minutes. Antioxidant activity was expressed as percent of inhibition relative to the control, using the following
3.5 Antioxidant activities of MSCE
Three antioxidant assays were used to evaluate the antioxidant activities on the MSCE with higher TFC value S7 and R1 selected from the RSM model and ramp function. Both methods (DPPH and FRAP) compared with the acid ascorbic (AA), and the β-carotene bleaching assay was compared with the BHT. In scavenging activity assay, the proton radical scavenging activity is known to be one of the several mechanisms for measuring antioxidant activity. DPPH contain proton free radical and its absorbance can be read at 517 nm. Generally, when DPPH encounters radical proton scavengers, its purple colour fade, and change to yellowish colour show present of antioxidant in the sample [48].
The FRAP assay measures the reducing potential of an antioxidant reacting with a ferric tripyridyltriazine complex and producing coloured ferrous tripyridyltriazine (blue colour) at low pH [49,50]. The present of the compounds can track by measured the increased the absorbance value as the compound started breaking the free radical by donating a hydrogen atom. The previous study shows that FRAP assay can be used to measure the mechanism of Fenton reaction by chelating metal ion such as Fe2+ which converts the hydrogen peroxide to hydroxyl radical [51].
In the β-carotene bleaching assay, linoleic acid was used to creates the hydroperoxides compound as free radicals during the incubation. This free radical was used to evaluate the antioxidant activity of MSCE to maintain the yellow colour of the test solution after 120 minutes [52]. The antioxidant activity from S7 is lower than acid ascorbic and BHT but similar to the R1. The similarity is probably because of the total flavonoid content are similar even though the parameter of the extraction condition is different. The present of oligomer procyanidin as in table 7 should yield high scavenging activity compared to the monomer such as epicatechin. A previous study was done by Vennat et al. (1994) [53] showed that monomer procyanidin is less effective against superoxide anion compare to the dimers and trimes procyanidin, while the heptramer and hexamers demonstrate greater superoxide scavenging activity than a tetramer.
Even though there is evidence on effectiveness oligomer procyanidin but the antioxidant activity was influenced by other factors such antioxidants concentration, method of extraction, extraction medium, pratical size of natural sources, temperature, Ph of medium [54], chemical structures and position in the molecule [55]. The oligomer procyanidin also presents in the cocoa bean, cocoa powder, and cocoa pod husk as mention before. The percentage of oligomer procyanidin is probably higher because of the extraction method involved ultrasound. The previous study shows that ultrasound can increase the procyanidin content in the extract compared to another method. The method extraction on larix gmelinii was compared between 80% solvent (ethanol) and ionic liquid such as Bmim-Br (1-butyl-3-methylimidazolium bromide) [56]. It was found that more 98% of the procyanidins content extracted during the first 30 minutes of UAE, and the result did not increase any further after 30 minutes as the application of ultrasound continues.
We conducted a kinetic study (data not published) for the S7 (concentration between 1000 – 8000 µg/ml) at the same wavelength used on antioxidant method. S7 antioxidant (DPPH and FRAP) are increased as the concentration increase with more than 50% scavenging effect and one µMoles/L respectively. For BCB assay, the MSCE S7 had a similar antioxidant activity to BHT as the concentration increase.
Table 8: Antioxidant activities of MCSE.
Antioxidant assay | S7 | R1 | Acid Ascorbic | BHT |
a) Scavenging activity assay (%) | 30.83±1.55 | 29.56±1.13 | 73.86±2.26 | n.d |
b) Ferric reducing/antioxidant power (µM Fe II g-1 extract) | 319.20±0.18 | 356.30±0.24 | 816.05±0.12 | n.d |
c) β-Carotene–linoleate bleaching assay (%) | 94.85±0.39 | 93.79±1.79 | n.d | 99.87±1.89 |
Point 2: Use either US or UK English
Response 2: the manuscript were written with British English style and was proofreading by a professional.
Point 3: Line 51: levels?
Response 3: Rephrase the word levels to the “yield.”
Point 4: Line 56: states? Kinds?
Response 4: adding an example of flavonoid “are flavan-3-ols and flavanols such as catechin and quercetin” and adding one more reference.
Point 5: Line 58: have different flavonoid yield?
Response 5: Rephrase word “flavonoid yield” to the “total flavonoid content.”
Point 6: Line 84: in analytical grade
Response 6: Edit according to the reviewer suggestion
Point 7: Line 121: what do you mean: “were tested for statistical data.”
Response 7: Statistical data refer to the p-value of the data collection in the simulator run. Since there might be confusion on the term, the word “for statistical data” is removed.
Point 8: Line 52: what do you mean: “the type polyphenols.”
Response 8: Rephrase “type of polyphenols” to the “composition of polyphenols.” The previous study only revealed the value of total phenolic content from the assay but did not reveal any composition of the polyphenols either it is flavonoid, phenolic acid, or other types of polyphenols.
Point 9: Line 22: what do you: “significant extraction”; extraction level?
Response 9: rephrase to “model was significant in the interaction between all variables.”
Point 10: Line 96: what do you: “was created a base on”
Response 10: Rephrase the words “ a base on” to the “according to”
Point 11: Figure 1 to 3
Response 11: More information and explanations were added such as the usage of mixture solvent (solvent –water), the decreasing TFC value after optimum time, and power consumption of the UAE as below:
Using solvent mixture could increase the amount to flavonoid or other compound composition extract from both ends of the polarity (highest and low polarity compound) [34]. By applying high temperature during the extraction process, the polarity of solvent can be decreased, and suitable for the targeted compound as the solvent mixture (water-solvent) could improve the effectiveness of extraction and increase surface area for solvent solid contact [35].
A previous study was done by Suslick (1990) [38] on sonochemistry; there is four variation of cavitation (acoustic, hydrodynamic, optic and particle) involve during the ultrasound irradiation. During the process, only acoustic and hydrodynamic cavitation can produce the intensities needed to induce the physical or chemical changes [39]. Even though the chemical influences of ultrasound may not happen direct interaction with molecular species but from the cavitation occurrence (formation, growth, and implosive microbubble or collapse cavities) may release a large amount of highly localised energy that can change the chemical in the system[40].
Furthermore, free radicals are formed during the acoustic cavitation, and at ambient temperature, these radicals could be utilised for acceleration of chemical reactions. Another factor such as local turbulence and acoustic streaming (liquid micro-circulation) might contribute the decreasing the mass transfer resistance in the system [40]. Therefore, ultrasound irradiation can be considered as a combination of a chemical reaction by utilising the factor mention associated with an increase in temperature[41]. All the factor mention before probable reasons why the TFC value decrease when it reaches an optimal value.
UAE offers several advantages compared to conventional extraction, for example, simplified manipulation, high reproducibility in shorter times, reduced solvent consumption and lower energy consumption. For comparison, a study was done by Lian He et al. (2018)[42] on extraction of polysaccharide from Dendrobium officinale stem show that UAE (20 kHz, at 400 W) give the highest yield 20.55% polysaccharide compare to the MAE (600 W) 17.74% and hot water extraction 14.77% polysaccharide. The other study was done by Carla and Andrea (2018)[43] on extraction polyphenols from saffron floral using MAE (800 W), and UAE (26 kHz, 200 W) obtain a similar result. By using UAE, polyphenols recovered from saffron floral is 4016 mg GAE 100 g DM-1 compare to the MAE 3108 mg GAE 100 h DM-1 [43]. Massive different between the method of extraction show that another method should increase their duration of the treatment (hence increase energy consumption) or increase solvent consumption to obtain a similar result to extraction using UAE. Thus the UAE can be optimised by using less energy consumption, shorten the duration of the extraction process and decrease solvent consumption.
Point 12: Line 168: a similar result was obtained by
Response 12: Edit according to the reviewer suggestion
Point 13: Line 183: do you mean: is not significantly different between…?
Response 13: Edit according to the reviewer suggestion
Point 14: Line 80: check the bracket
Response 14: Remove the bracket word (PPPK) to avoid the confusing.
Reviewer 2 Report
The work presented by authors is very interesting, however the manuscript must to be improved.
In introduction authors could add any information abour the ultrasoud assisted extraction in order to explain how this alternative technology is used in industries to increase extraction rates using less energy than other extraction methods.
Material and methods:
On 2.2. Ultrasound-assisted extraction. Authors must to add information concerning power of ultrasounds (W), agitation speed provided by a mixer (mechanical or magnetic agitator) adn the particle size of the obtained samples.
In order to compare the effect of ultrasounds authors must to provide information about solid-liquid extraction with fixed parameters (solvent composition, temperature and agitation speed). Abscence of this information makes difficult to conclude about the efficiency of ultrasounds during extraction.
On 2.3. RSM design and statistical analysis. Why authors makes the choice of ethanol concentrations between 63.18 and 96.82 %?, Can authors add some litterature to validate this choice?. Can authors produce any economy in ethanol use, it should be interesting think what could happend if an industrial compony decide to use this information.
Conclusions:
line 225: How authors can conclude that "...UAE was used to enhance the extraction of flavonoids compounds...", if you do not shown a comparison with a classical extraction without ultrasounds.
Author Response
Response to Reviewer 2 Comments
Point 1: In introduction, authors could add any information about the ultrasound assisted extraction in order to explain how this alternative technology is used in industries to increase extraction rates using less energy than other extraction methods.
Response 1: There are no references found regarding the energy input used by the machine. From experience in the extraction process at the Malaysia Cocoa Board, the energy input was basically in between 50 – 400 W base on the frequency of the selected devices. Usually, the ultrasound devices can only operate one frequency either 20 kHz or 40 kHz (most of the reference found). Some other devices can operate any frequency by changed its transducer. The explanation of the UAE mechanism is included as below:
The principle of UAE is based on fundamental of wave frequency different than fundamental of sound. Generally, sound waves classified into three category which are audible waves (10 Hz – 20 kHz), infrasonic waves (<16 Hz) and ultrasonic waves (between 20 kHz and 10 MHz) [25]. Above than 10 MHz are categories as a microwave. Usually, the power use in UAE are in between 50 – 400 W. During the UAE process, the microbubbles were generated due to the different amplitude of ultrasonic[26]. In result of continuous changes of amplitude, the cavitation bubbles oscillate and collapse, create several physical effects. Some physical effects included shock waves, microjets, turbulence, and shear force in result improve the mass and heat transfer through the disruption of cell walls and increase pores on the surfaces of the cell leading to the improved release of the target bioactive compounds from the natural sources [27].
Point 2: On 2.2. Ultrasound-assisted extraction. Authors must add information concerning power of ultrasounds (W), agitation speed provided by a mixer (mechanical or magnetic agitator) and the particle size of the obtained samples.
Response 2: The experiment was done by using an ultrasound cleaner (sonicate cleaner) as mention in the manuscript. There is no agitation speed or type of agitator were mentioned in the manual of the machine, only the frequency 40 kHz with the energy input 296 W (adding in the manuscripts). The transducer was located at the bottom of the tank wall. The particle size was not evaluated during the experiment.
Point 3: In order to compare the effect of ultrasounds authors must to provide information about solid-liquid extraction with fixed parameters (solvent composition, temperature and agitation speed). Absence of this information makes difficult to conclude about the efficiency of ultrasounds during extraction.
Response 3: There is no experiment was conducted to evaluate the effectiveness of the ultrasound by comparing with the experiment without ultrasound as a request by the reviewer. However, the information on the previous study was included in the manuscripts on mixture solvent (solvent –water), the decreasing TFC value after optimum time, and power consumption of the UAE as below:
Using solvent mixture could increase the amount to flavonoid or other compound composition extract from both ends of the polarity (highest and low polarity compound) [34]. By applying high temperature during the extraction process, the polarity of solvent can be decreased, and suitable for the targeted compound as the solvent mixture (water-solvent) could improve the effectiveness of extraction and increase surface area for solvent solid contact [35].
A previous study was done by Suslick (1990) [38] on sonochemistry; there is four variation of cavitation (acoustic, hydrodynamic, optic and particle) involve during the ultrasound irradiation. During the process, only acoustic and hydrodynamic cavitation can produce the intensities needed to induce the physical or chemical changes [39]. Even though the chemical influences of ultrasound may not happen direct interaction with molecular species but from the cavitation occurrence (formation, growth, and implosive microbubble or collapse cavities) may release a large amount of highly localised energy that can change the chemical in the system[40].
Furthermore, free radicals are formed during the acoustic cavitation, and at ambient temperature, these radicals could be utilised for acceleration of chemical reactions. Another factor such as local turbulence and acoustic streaming (liquid micro-circulation) might contribute the decreasing the mass transfer resistance in the system [40]. Therefore, ultrasound irradiation can be considered as a combination of a chemical reaction by utilising the factor mention associated with an increase in temperature[41]. All the factor mention before probable reasons why the TFC value decrease when it reaches an optimal value.
UAE offers several advantages compared to conventional extraction, for example, simplified manipulation, high reproducibility in shorter times, reduced solvent consumption and lower energy consumption. For comparison, a study was done by Lian He et al. (2018)[42] on extraction of polysaccharide from Dendrobium officinale stem show that UAE (20 kHz, at 400 W) give the highest yield 20.55% polysaccharide compare to the MAE (600 W) 17.74% and hot water extraction 14.77% polysaccharide. The other study was done by Carla and Andrea (2018)[43] on extraction polyphenols from saffron floral using MAE (800 W), and UAE (26 kHz, 200 W) obtain a similar result. By using UAE, polyphenols recovered from saffron floral is 4016 mg GAE 100 g DM-1 compare to the MAE 3108 mg GAE 100 h DM-1 [43]. Massive different between the method of extraction show that another method should increase their duration of the treatment (hence increase energy consumption) or increase solvent consumption to obtain a similar result to extraction using UAE. Thus the UAE can be optimised by using less energy consumption, shorten the duration of the extraction process and decrease solvent consumption.
Point 4: On 2.3. RSM design and statistical analysis. Why authors makes the choice of ethanol concentrations between 63.18 and 96.82 %?, Can authors add some litterature to validate this choice?. Can authors produce any economy in ethanol use, it should be interesting think what could happend if an industrial compony decide to use this information.
Respose 4: The RSM was designed according to the Centre Composite Design (CCD) as mention in the manuscripts. There is five level in the CCD compare to the Box-Behnken design (BBD) only have three levels. The software set the alpha value to the constant 1.68179 with k<6 (k= number of variables), so we can have a rotatable design for the model. The rotatable design means the model can be simulated outside of the model box and extend to the maximum value of alpha in between (1 – 1.68179). There is a formula calculation on how to get the alpha value, but do not discuss in the manuscripts as the software can calculate the alpha value.
Point 5: line 225: How authors can conclude that "...UAE was used to enhance the extraction of flavonoids compounds...", if you do not shown a comparison with a classical extraction without ultrasounds.
Response 5: There is no experiment were conducted to evaluate the effectiveness of the ultrasound, so rephrase the sentences without word “enhancement.”
Round 2
Reviewer 1 Report
this version is better. The quality of Fig4-5 can be improved if possible, in a more professional way.